# Photoinactivation of *Pseudomonas aeruginosa* Biofilm by Dicationic Diaryl-Porphyrin

**DOI:** 10.3390/ijms22136808

**Published:** 2021-06-24

**Authors:** Viviana Teresa Orlandi, Eleonora Martegani, Fabrizio Bolognese, Nicola Trivellin, Francesco Garzotto, Enrico Caruso

**Affiliations:** 1Department of Biotechnologies and Life Sciences, University of Insubria, Via JH Dunant 3, 21100 Varese, Italy; e.martegani@uninsubria.it (E.M.); fabrizio.bolognese@uninsubria.it (F.B.); enrico.caruso@uninsubria.it (E.C.); 2Department of Industrial Engineering, University of Padova, Via Gradenigo 6A, 35131 Padova, Italy; nicola.trivellin@unipd.it; 3Institute of Oncology IOV—IRCCS, 35128 Padova, Italy; f.garzotto@gmail.com

**Keywords:** diaryl-porphyrins, antimicrobial photodynamic therapy, aPDT, *Pseudomonas aeruginosa*, biofilm

## Abstract

In recent years, antimicrobial photodynamic therapy (aPDT) has received increasing attention as a promising tool aimed at both treating microbial infections and sanitizing environments. Since biofilm formation on biological and inert surfaces makes difficult the eradication of bacterial communities, further studies are needed to investigate such tricky issue. In this work, a panel of 13 diaryl-porphyrins (neutral, mono- and di-cationic) was taken in consideration to photoinactivate *Pseudomonas aeruginosa*. Among cationic photosensitizers (PSs) able to efficiently bind cells, in this study two dicationic showed to be intrinsically toxic and were ruled out by further investigations. In particular, the dicationic porphyrin (P11) that was not toxic, showed a better photoinactivation rate than monocationic in suspended cells. Furthermore, it was very efficient in inhibiting the biofilms produced by the model microorganism *Pseudomonas aeruginosa* PAO1 and by clinical strains derived from urinary tract infection and cystic fibrosis patients. Since *P. aeruginosa* represents a target very difficult to inactivate, this study confirms the potential of dicationic diaryl-porphyrins as photo-activated antimicrobials in different applicative fields, from clinical to environmental ones.

## 1. Introduction

In recent years, in clinical field, the attention was focused on “ESKAPE” pathogens for their ability to “escape” from antimicrobials’ action. The term “ESKAPE” arises from the following six name species: *Enterococcus faecium*, *Staphylococcus aureus*, *Klebsiella pneumoniae*, *Acinetobacter baumannii*, *Pseudomonas aeruginosa* and *Enterobacter* spp [1]. These bacterial species are associated with most of nosocomial infections and the highest risk of mortality [2]. All of them have been recently reported by the World Health Organization (WHO) in the list of the 12 bacterial species against which new antibiotics are urgently needed [3]. In particular, *P. aeruginosa*, thanks to its wide genome (5–7 Mbp), displays a wide capacity to use various carbon sources and adapts to several environments including soils, waters, sewages and is a common part of the microflora of different animals. In addition, the highest part of its genome is dedicated to regulatory genes and networks that are fundamental for the response and adaptation to different and changing environments [4]. In humans, it causes severe acute or chronic infections in a variety of tissues and body sites, including skin, middle-ear, eyes and urinary tract, especially in immunocompromised patients [5,6]. *P. aeruginosa* is a leading cause of nosocomial infections, including, i.e.**,** urinary tract catheter-associated infections (CAUTIs), central-line associated bloodstream infections (CLABSIs) and ventilator-associated pneumonia (VAP), as well as chronic lung infections in cystic fibrosis patients [7,8]. The major cause of persistent *P. aeruginosa* infections is the presence of biofilm, formed on tissues or on the surface of surgical implants or medical devices. *P. aeruginosa* cells attach on a surface through twitching motility driven by type IV pili and develop microcolonies. At this step, they secrete exopolysaccharides (Psl, Pel and alginate) and other components such as polypeptides and extracellular DNA. During biofilm maturation, the community acquires its typical three-dimensional structure that planktonic cells can leave for colonizing other surfaces [9]. As a whole, biofilm formation and production of many virulence factors (i.e.**,** pyocyanin, pyoverdine, elastases, proteases, rhamnolipids, exotoxin A) contribute to pathogenicity in *P. aeruginosa* [10]. Furthermore, *P. aeruginosa* is characterized by an outer membrane that acts as a selective barrier to prevent the antibiotic entrance, in addition to several non-specific porins that govern membrane permeability [11]. In addition, a wide variety of efflux pumps (i.e.**,** mexAB-OprM, MexCD-OprJ, MexEF-OprN) are responsible for the resistance to different classes of antibiotics (β-lactams, quinolones and aminoglycosides) [12]. The production of extended-spectrum β-lactamases and enzymes modifying aminoglycosides worsen the spread of multidrug resistant strains [13].

In this alarming scenario, the visible-light based techniques are gathering attention. The main approach exploiting visible light as an anti-infective agent is antimicrobial photodynamic therapy (aPDT) [14]. The photodynamic process is based on the simultaneous presence of three components: a source of light energy, a photosensitive compound and molecular oxygen. Upon the photoexcitation of a photosensitizer (PS) with an appropriate light wavelength, several reactive oxygen species (ROSs) are released: hydroxyl radical (OH^.^), hydrogen peroxide (H_2_O_2_), superoxide anion (O_2_^-^) and singlet oxygen (^1^O_2_). The elicited oxidative stress compromises the integrity of macromolecules, including lipids, proteins and nucleic acids, and cellular structures leading to microbial cell death [15,16]. An interesting advantage of this technique is based on the efficacy observed against both sensitive and antibiotic resistant strains [17]. In addition, several studies highlight the potential of aPDT on microbial biofilms [18,19,20,21]. Until now, several photosensitizers have been taken into consideration for the inhibition and/or eradication of *P. aeruginosa* biofilms. Methylene blue, belonging to phenothiazine compounds, was successful as antibiofilm PS [22]. Curcumin, a well-known natural PS, inhibited the biofilm formation of *P. aeruginosa*, reducing the EPS (extracellular polymeric substance) production by 94% [19]. A PS belonging to the boron-dipyrro-methene (BODIPY) class, upon activation with green light, was efficient not only in inhibiting, but also in eradicating biofilms of *P. aeruginosa* PAO1 [23]. Among the most investigated dyes in the photodynamic field, the family of porphyrins showed a promising potential in biofilm treatment. For example, the 5,10,15,20-tetrakis [4 -(3-N,N-dimethylammoniumpropoxy)phenyl]porphyrin (TAPP) inhibited the biofilm formation of *P. aeruginosa* [24]. A recent study showed the efficacy of cationic zinc-porphyrins in disrupting and detach the matrix of 16–18 h-old biofilms of *P. aeruginosa* [25]. Porphyrins are widely distributed in nature in both prokaryotic and eukaryotic organisms as components of cytochromes, heme groups and chlorophylls, and are involved in many biological processes, such as photosynthesis and oxygen or electron transport [26]. The extensive electron delocalization on the macrocycle ring is responsible for the intense absorption of porphyrins in the visible range. Their typical spectrum is characterized by a higher absorption band around 420 nm, known as Soret band, and weaker bands between 500 and 600 nm (Q bands), making porphyrins suitable PSs to be activated by different light sources, including wide-spectrum emission lamps, sunlight, light-emitting diodes (LEDs) and lasers [27]. Along with this aspect, porphyrins show strong photosensitizing abilities, due to their long-lived triplet state and notable yield of singlet oxygen production, making these compounds almost ideal photosensitizers [28]. Further, the chemical synthesis of porphyrins is relatively simple and cost-efficient and generally involves the condensation of pyrroles with suitable aldehydes. The resulting tetra-pyrrolic ring is a versatile skeleton bearing four different substituents in meso-positions, meaning that high number of combinations could lead to the production of different molecules with desired chemico-physical features [27]. Photosensitizers should not be toxic and mutagenic in the dark towards both eukaryotic and prokaryotic cells. In literature, conflicting observations have been reported on dark toxicity of the most investigated porphyrin, tetracationic TMPyP (5,10,15,20-Tetrakis(1-methyl-4-pyridinio)-porphyrin tetra(p-toluenesulfonate). Eckl assumed that the dark toxicity of TMPyP, observed in several bacterial species, could be attributed to photoinactivation effects caused by any residual light in a laboratory. When keeping the bacteria under dark conditions (<10 nW cm²), no dark toxicity in *Escherichia coli* was detected for a very high concentration of TMPyP (250 µM) and incubation times up to 24 h [29]. On the other hand, *P. aeruginosa* strains isolated from Fibrosis Cystic patients were sensitive in the dark to porphyrins [30].

Among synthetic porphyrins, diaryl-porphyrins, bearing two substituents in two meso-positions, were shown to be efficient PSs both in antitumoral and in antifungal applications [31,32]. In the present study a panel of 5,15 meso-substituted diaryl-porphyrins was assayed for anti-biofilm activity. The compounds were chosen due to their different degree of amphiphilicity, molecular symmetry and charge and were tested against *P. aeruginosa*.

## 2. Results

### 2.1. Panel of Diaryl-Porphyrins

A panel of novel diaryl-porphyrins, synthetized by our group and previously tested as antimicrobials [33] antifungals [32] and antitumorals [31,34], was investigated for anti-Pseudomonas activity (Table 1). The neutral and asymmetrical P1 and P2 bear in meso-positions (positions 5 and 15) a pentafluorophenyl group, associated with a C4 or C8 para-bromoalkyloxy-phenyl group, respectively. In compounds P3 and P5, the C4 or C8 bromoalkyloxy-phenyl chain is associated with a phenyl group in position 5. The symmetrical P4 and P6 bear two phenyl groups with para-bromobutoxy and two para-bromooctanoxy chains, respectively. The positive charge of cationic diaryl-porphyrins derives from a pyridinium group. All monocationic PSs are asymmetrical molecules bearing a phenyl (P7, P8) or a pentafluorophenyl group (P9, P10) in position 5 and a pyridinobutoxy-phenyl (P7, P9) or pyridinooctanoxy-phenyl group (P8, P10) in position 15. The three dicationic symmetric porphyrins are characterised by benzyl group as alkylating group of the pyridyl substituent (P11), or alkoxy-linked pyridinium at the end of four (P12) or eight (P13) carbon chains.

#### Intrinsic Toxicity of Diaryl-Porphyrins

The effect of diaryl-porphyrins under dark incubation was investigated because the “ideal” PS should not display intrinsic toxicity. The reduction of microbial viability in the presence of photosensitizer without irradiation should be avoided [35]. Since PSs are dissolved in DMSO, it was evaluated that, under the tested conditions, the solvent did not impair microbial growth (Figure 1). Indeed, *P. aeruginosa* was almost insensitive to neutral and monocationic porphyrins up to the longest tested dark incubation (6 h). On the other hand, two of the three dicationic PSs were intrinsically toxic. The observed killing effects increase with longer incubation times: after 6 h of dark incubation, P12 impaired the growth of sample at ~10^4^ CFU/spot and P13 at ~10^3^ CFU/spot (Figure 1).

### 2.2. Diaryl-Porphyrins Binding and Photoinactivation Rates

As an optimal interaction between PS and microorganism is required for the following oxidative stress elicited by irradiation [36], the binding rate of porphyrins to microbial cells was evaluated. The yield of PS binding was strictly related to diaryl-porphyrin charge. After 1 h of dark incubation, neutral PSs (P1–P6) showed a very low affinity, less than ~8% of PS binding except P2 with a binding yield of ~15% (Figure 2). On the other hand, positively charged compounds (P7–P13), both mono- and dicationic, were able to strongly interact with *P. aeruginosa* cells (Figure 2).

Since the dark toxicity is not desired in aPDT applications, P12 and P13 were excluded from the following investigations. The antimicrobial potential of the remaining diaryl-porphyrins was investigated under irradiation of a LED emitting at 410 nm and fitting with typical Soret band of porphyrins. Since *P. aeruginosa* is sensitive to light at 410 nm [37], a light dose without toxic effect (20 J/cm^2^) was chosen for the activation of porphyrins. After 1 h of dark incubation with diaryl-porphyrins to favor the interaction between PS and cell, bacteria were irradiated. The administration of neutral porphyrins (P1–P6) to *P. aeruginosa* did not affect cell viability upon activation by blue light, neither at the lowest cell concentration (10^2^ CFU/spot) (Figure 3).

Higher concentrations (up 30 µM) of neutral PS failed in photoactivation (data not shown). Among the four monocationic porphyrins, P7 10 µM caused a Log10 reduction of ~2.5 unit. It is noteworthy that longer chain (8 carbon) in P8 or the presence of 5 Fluoro atoms on phenyl residue in 5 positions in P9, compromised completely the photoactivation. The dicationic P11 displayed an activity twofold higher than monocationic P7 (Figure 3). Thus, at the end of this screening, the dicationic diaryl-porphyrin P11 resulted the best candidate to photoinactivate the suspended form of *P. aeruginosa*: a significant decrease of more than 4 Log units (from 10^8^ to 10^4^ CFU/mL) was obtained upon PDT treatment (Figure 4).

### 2.3. Photodynamic-Inhibition of Biofilm Formation

The ability to form structured communities, both on inert surfaces and biological tissues, renders *P. aeruginosa* particularly tolerant to conventional antibiotic therapies. Photodynamic therapy (PDT) is a promising approach to tackle bacterial infections in biofilm lifestyle, both in inhibiting biofilm formation and/or eradicating formed biofilms. Notwithstanding, few studies have been carried out employing porphyrins for the photodynamic treatment of microbial biofilms, and moreover no reports describe the inhibition of *P. aeruginosa* biofilms by photoactivation of porphyrins. Thus, the potential of P11 in inhibiting the formation of biofilms was tested on PAO1 strain and two clinical isolates, *P. aeruginosa* UR48 isolated from a patient with catheter-associated urinary tract infections (CAUTI), and BT1 from the sputum of a cystic fibrosis (CF) patient [38,39]. Among the chosen strains, BT1 formed a biofilm with the highest biomass value (OD590~27), while PAO1 and UR48, 7 and 12, respectively (Figure 5). Since cells forming the adherent phase of all the considered strains showed a comparable density (~10^8^ CFU/well), it can be inferred that CF isolate was able to hyperproduce extracellular components of matrix biofilm. Upon administration of DMSO 2.5%, and even more P11 30 µM dissolved in DMSO, an increase of crystal violet staining was observed in PAO1. Since no changes in cellular concentrations of planktonic and adherent phases were observed, DMSO could induce the formation of extracellular matrix. However, in all the strains, the combination of P11 and blue light irradiation (30 J/cm^2^) caused a relevant and statistically significant inhibition of biomass adhesion. Similarly, the cellular concentrations of planktonic and adherent subpopulations were significantly lower than control samples (Figure 5).

The photoinactivation protocol was also applied to *P. aeruginosa* PAO1 strain expressing the green fluorescent protein (GFP) under arabinose induction. Before confocal laser scanning microscope (CLSM) analysis, GFP expression was induced by arabinose to highlight viable cells with functional and active protein machinery. In the control biofilms the detection of GFP signal can be appreciated and no signal was observed under the combination of photosensitizer and light at 410 nm (Figure 6). The obtained results support the effectiveness of PDI mediated by diaryl-porphyrins in inhibiting biofilm formation.

### 2.4. Photodynamic-Eradication of Formed Biofilms

If biofilm inhibition is a crucial step in infection prevention, the control of infection through eradication of mature biofilm remains the most arduous challenge. Therefore, the effect of porphyrin-mediated photodynamic treatment was evaluated on 24 h-old biofilms formed by the model strain PAO1. PDT experimental conditions were set as follows: upon 24 h of biofilm growth, porphyrin P11 (30 µM final concentration) was gently administered to the samples, without modifying the biofilm environment and, after 1 h of dark incubation, samples were irradiated with light at 410 nm (30 J/cm^2^). DMSO-treated samples and dark controls were included in each experiment. The total adherent biomass did not significantly change upon DMSO or P11 administration, in both light and dark conditions, as compared to the untreated dark sample. Interestingly, activation of P11 by blue light caused a significant decrease of 2 Log units in both adherent and planktonic populations (Figure 7B,C). These results suggested a mild anti-biofilm effect on the cellular component of biofilm, both sessile and planktonic, of the dicationic diaryl-poprhyrin P11.

On the other hand, the photoinactivation protocol (P11 30 µM, 30 J/cm^2^) applied to a 24 h grown biofilm of *P. aeruginosa* PAO1 expressing GFP showed a clear antimicrobial effect. In the control biofilms, a comparable fluorescent signal was detected (Figure 8A–E), and in the photoinactivated biofilm the fluorescent signal almost disappeared (Figure 8F). It can be hypothesized that the treatment impaired cell functions, including the activity of the cellular protein synthetic machinery. Even if the viability of sessile bacteria was slightly compromised, most of the cells seemed to be damaged immediately after PDT treatment.

## 3. Discussion

Since *P. aeruginosa* is a pathogen difficult to eradicate for its resistance to antibiotics and tolerance to antimicrobial treatments, it is very interesting to acquire information on its sensitivity to novel drugs, independently from photodynamic applications. The dicationic compounds showed different activities: the porphyrins, P12 and P13, bearing substituents with longer chains (4 carbon and 8 carbon) in 5 and 15 positions were toxic, while P11 bearing a benzyl chain in the same positions was not intrinsically toxic. Therefore, the presence of an alkyl chain that increases the degree of lipophilicity can favour the cross and perhaps the injury of the outer membrane that represents the ideal target for new approaches [40].

In literature, conflicting observations have been reported on dark toxicity of the most investigated porphyrins [29,30]. It is noteworthy that P12 and P13 show a higher Soret band than the other diaryl-porphyrins under investigation in this study (Figure 9), and it cannot be ruled out that the occasional exposure to wide spectrum daylight for experimental set-up could be sufficient to elicit a mild photo-oxidative stress.

It can be hypothesized that the positive charge on P11 is strongly delocalized, in fact the pyridine nucleus is directly coordinated with the tetrapirrole system of porphyrin. The greater charge distribution should disadvantage a strong interaction of diaryl-porphyrins with anionic counterparts of the cell wall of both Gram-negative and Gram-positive bacteria. Instead, P12 and P13 show a strongly localized charge only in the pyridine positioned far from the tetrapyrrole nucleus, this may suggest a strong electrostatic interaction with the bacterial cell wall. The electrostatic interaction may be sufficient to disassemble cell wall integrity, impairing bacterial viability, independently from irradiation.

Moreover, the eukaryotic microorganism *Candida albicans* showed to be sensitive in the dark to most cationic and dicationic diaryl-porphyrins tested in this study, irrespective of Soret band height [32]. Cormick reported that tri- and tetracationic porphyrins were more tightly bound to *C. albicans* cells than anionic porphyrins, supporting the requirement of positive charge on porphyrins to promote the electrostatic interaction with the yeast cell wall [41]. Indeed, the net negative charge of yeast cell wall is conferred by a robust polysaccharide skeleton linked to mannoproteins and chitin [42]. Since diaryl-porphyrins were intrinsically toxic to *P. aeruginosa* and *C. albicans* in a different rate, irrespective of their absorbance spectrum, their potential activation by indoor light cannot be considered as the only factor appointed for intrinsic toxicity.

On the basis of the obtained results, several considerations may be carried out about efficiency of binding and photo-inactivation yields. Neutral diaryl-porphyrins were not able to bind tightly to *P. aeruginosa* cells, as the most part of PSs was recovered in the supernatant, and furthermore, upon irradiation, no killing was observed. On the other hand, cationic compounds (P8, P9, P10), even if tightly bound to the cell wall, did not elicit any photooxidative stress upon irradiation. Thus, a good binding is not necessarily the only requirement for a successful photoactivation process in *P. aeruginosa*. The factor that could affect the potential activity of PSs is the capacity of PSs to bind and penetrate the cell wall and, possibly, reach cytoplasmic targets. In this regard, Sulek reported that cationic TMPyP was not efficient in photoinactivating *E. coli* cells, and it was necessary to administer verapamil, an efflux pump inhibitor, to potentiate the antibacterial effect [43]. The inhibitor could increase PS’s accumulation in the bacteria cell of molecules attached to cell wall via electrostatic interactions. A damage of the outer surface of the outer membrane of Gram-negative bacteria could be less dangerous than that of the inner part of the same envelope or the cytoplasmic environment. The photoactivation of P11 impaired the cell wall of *E. coli*: the outer membrane of approx. 90% of the treated cells appeared fuzzier and lacked the pronounced margin of the envelope of control cells, and 60–70% of the cells underwent envelope and cytoplasmic changes with the arising of honeycomb-like structures [33]. Thus, the impairment of the cell wall seems relevant and necessary upon photodynamic treatment. The outer membrane forms a barrier that impairs neutral PS binding and penetration. The high yield of binding of cationic diaryl-porphyrins (80–100%) could be ascribable to the electrostatic force displayed between negative-charged lipopolysaccharides on the outer layer of the outer membrane and positively charged PSs. As previously reported by other authors, one or more positive charges are required on the PS structure for a good interaction with Gram-negative bacteria [43]. The mechanism of binding of cationic PSs with bacteria is the so-called “self-promoted uptake” pathway. This pathway involves the binding of the cationic molecules to LPS that results in the progressive displacement of divalent cations (Ca^2+^, Mg^2+^) electrostatically bound to the LPS, thereby weakening the outer membrane. The destabilization of the LPS coat results in the formation of “cracks” in the permeability barrier, and divalent cations neutralize the negative character of cell membrane and eliminate electrostatic repulsion between PS and the bacterial envelope [43]. Furthermore, a partial hydrophobic feature associated to C4 and C8 substituents of P12 and P13 could facilitate the cross of the cytoplasmic membrane. A certain degree of amphiphilicity was reported as peculiar and advantageous for porphyrins in photodynamic applications [43]. Neutral PSs that do not interact with *P. aeruginosa*, neither affect its viability upon dark incubation.

As reported in literature, it was necessary to employ high concentration of TMPyP (225 µM) to induce 4-fold reduction in *P. aeruginosa* biofilms and cause detachment of the biofilm from the substratum [44]. Another PDI study using a lower concentration of TMPyP (20 µM) under 64.8 J/cm^2^ demonstrated that in *P. aeruginosa* polysaccharides of the biofilm matrix may be a primary target of photodynamic damage [45]. Patel et al. reported that a cationic zinc (II) porphyrin, ZnPor, at low concentration (~20 µM), resulted in the extensive disruption and detachment of the matrix of 16–18 h-old biofilms of *P. aeruginosa* [25]. However, it has been recently hypothesized that the presence of negative charges in the EPS matrix could protect bacteria from the interaction with positively charged PSs. Furthermore, larger molecules are disadvantaged in the penetration through the biofilm matrix if compared to smaller ones [46]. Since no standard procedures are yet available for biofilm studies as guidelines for MIC evaluation value (minimal inhibitory concentration) [47], comparisons can not be made between different experimental approaches. In our case, stringent conditions were applied in antibiofilm PDT and no changes in biofilm environment were made upon biofilm growth. The diaryl-porphyrin P11 (30 µM) caused a 2-Log unit depletion of both adherent and planktonic populations of PAO1 biofilm. In addition, confocal analyses allowed to observe the biofilm adherent population immediately upon photo-treatment. PDT provoked a certain damage to embedded PAO1 cells, suggesting that they could be more sensitive to other antimicrobial agents delivered upon photodynamic treatment. The embedding of porphyrins in nanoparticles or polymeric coatings could be exploited both for skin disinfection and surface sanitization [48,49].

In conclusion, this study confirmed that the dicationic diarylic porphyrin P11 that previously showed to be efficient in inhibiting both Gram-negative *E. coli* and *P. aeruginosa* and Gram-positive *Enterococcus faecalis* and *S. aureus*, is optimal to inhibit the formation of *P. aeruginosa* biofilm. Since a mild effect on formed biofilm was obtained, these results could pave the way through the development of combined antibiofilm strategies where P11-mediated PDT in addition to other antimicrobial approaches could successfully eradicate *P. aeruginosa* biofilms.

## 4. Materials and Methods

### 4.1. Photosensitizers

A panel of 13 diaryl-porphyrins previously described [31,33,34] has been used in this study (Table 1). As shown in Figure 1, P1–P6 porphyrins are neutral molecules, P7–P10 are monocationic and P11–P13 di-cationic, respectively. PSs were dissolved in DMSO (Sigma Aldrich, Milano, Italy) at a final concentration of 1 or 0.5 mM, as requested, and stored at 4 °C until needed.

### 4.2. Microbial Strains and Culture Conditions

Three different strains of *P. aeruginosa* were considered for this study: *P. aeruginosa* PAO1 as model pathogen to photoinactivate [4] and two clinical strains previously considered: UR48 from urinary tract infection [39] and BT1 from cystic fibrosis patient [38]. *P. aeruginosa* strains were grown in Luria Bertani (LB) (Sigma Aldrich, Milano, Italy) on an orbital shaker at 200 rpm, or in solid media (15 g/L agar) at 37 °C. For biofilm formation, *P. aeruginosa* strains were grown in M9 minimal medium added with glucose (10 mM) (Sigma Aldrich, Milano, Italy) and casamino acids (0.2% *V*/*V*) (Sigma Aldrich, Milano, Italy) at 37 °C in static conditions. When necessary, the bacterial concentration was determined by viable count technique. Briefly, an aliquot of each sample was ten-fold serially diluted and a 10 µL of each diluted and undiluted sample was inoculated on LB Agar. After overnight incubation at 37 °C, the colony count was performed and the corresponding cellular concentration expressed as CFU/mL was calculated.

### 4.3. Light Source

The lighting unit device is equipped with a head composed by 25 high power LEDs with maximum emission peak at 410 nm blue light, suitable for the activation of porphyrins and allows the uniform irradiation of a square area of 75 mm × 75 mm. The system is powered by a specific PC based control system, which allows the setting of irradiation time and irradiance values for a precise evaluation of the radiation fluence rate.

### 4.4. Photo-Spot Test Assay

The spot test previously optimized [50] was used to screen the intrinsic toxicity and the photoactivity of diaryl-porphyrins. Upon overnight growth in LB, *P. aeruginosa* PAO1 culture (~10^9^ CFU/mL) was suspended in phosphate buffer saline (PBS-KH_2_PO_4_/K_2_HPO_4_ 10 mM, pH 7.4) and 10-fold serially diluted from ~10^9^ to ~10^4^ CFU/mL in 96-well plates.

To investigate the intrinsic toxicity of PSs, undiluted and diluted bacterial suspensions were incubated in the dark with PSs 10 µM. After 10 min, 1 h or 6 h of dark incubation, volumes of ~5 µL of each sample were replica plated on LB agar. Untreated samples and DMSO treated samples were included as controls. After O/N incubation at 37 °C, the growth of treated samples was compared to control growth spots of decreasing cell density (from ~10^7^ to ~10^2^ CFU/spot, respectively). For example, if the growth spot at 10^2^ CFU/spot was not observed, a 2-log unit decrease was recorded. Similarly, a 3-log unit was recorded if the spot at 10^3^ CFU/spot was not observed. Thus, higher values correspond to higher dark toxicity. The experiments have been repeated at last three times with independent cultures.

To investigate the photo-inactivation rates of diaryl-porphyrins, the PSs were administered at a final concentration of 10 μM to samples of *P. aeruginosa* PAO1 at decreasing concentrations (from 10^9^ up 10^4^ CFU/mL), as previously described. After 1 h dark incubation, volumes of ~5 µL of each sample were replica plated on LB agar and irradiated under 410 nm light (20 J/cm^2^). After O/N incubation at 37 °C, the growth spot was checked and compared to untreated control of decreasing cell density (from ~10^7^ to ~10^2^ CFU/spot, respectively). For example, if the growth spot at 10^2^ CFU/spot was not observed, a 2-log unit decrease was recorded. Similarly, a 3-log unit was recorded if the spot at 10^3^ CFU/spot was not observed. Thus, higher values correspond to higher antimicrobial efficiency. Photo-spot tests were performed at least in triplicate.

### 4.5. Photoinactivation of Suspended Cells

Upon overnight growth of *P. aeruginosa* PAO1, cells were ten-fold diluted in sterile deionized water, to reach approximate concentrations of 10^8^ CFU/mL. Porphyrins were added to cell suspension at a final concentration of 10 µM. Untreated cells, DMSO-treated cells and not irradiated controls were also included. Cells were incubated in the dark for 60 min and then irradiated (20 J/cm^2^). Soon after irradiation, the number of viable cells was evaluated by viability count, as previously described. Photoinactivation experiments were performed at least in triplicate.

### 4.6. Photodynamic Treatment of Biofilms

The effect of diaryl-porphyrins in inhibiting the biofilm formation of *P. aeruginosa* was evaluated as follows. Overnight cultures of *P. aeruginosa* PAO1, UR48 and BT1 strains were diluted 500-fold in M9 minimal medium added with glucose (10 mM) and casamino acids (0.2% *V*/*V*) reaching a concentration of ~10^7^ CFU/mL and inoculated in 12-well microplate. Porphyrins were added at a final concentration of 30 µM and incubated in the dark for 1 h. Upon irradiation with a final dose of 30 J/cm^2^ (100 mW/cm^2^, 300 s), bacteria were grown O/N at 37 °C to form biofilm. In order to evaluate the effect of the different treatments on the cellular viability of suspended and adherent populations, the planktonic phase was axenically collected and adherent cells were recovered by scraping and suspended in 1 mL of PBS. Viable counts—expressed as CFU/mL in cell suspensions and as CFU/well in adherent biomass were estimated by a plate count technique, as previously described. The total adherent biomass was quantified by crystal violet (CV) staining. Briefly, planktonic biomass was removed and wells were washed once with 1 mL PBS. One millilitre of 0.1% (*W*/*V*) CV was added to each well for approximately 20 min to stain the biofilm, after which the CV was removed and each well was gently washed with 1 mL PBS. The remaining CV, which indicated the amount of biofilm present, was dissolved in acetic acid 30% for 10 min. The amount of solubilized dye was spectrophotometrically measured at 590 nm.

To evaluate the eradication of biofilm, overnight cultures of *P. aeruginosa* PAO1 were diluted 500-fold in M9 minimal medium added with glucose (10 mM) and casamino acids (0.2% *V*/*V*) reaching a concentration of ~10^7^ CFU/mL and inoculated in 12-well microplate to let form biofilm. 24 h-old biofilms were treated with PS at a concentration of 30 µM, dark incubated for 1 h and irradiated (30 J/cm^2^). After irradiation, the adherent biomass (OD_590_) and viable counts from sessile (CFU/well) and planktonic phases (CFU/mL), respectively, have been evaluated as previously described.

In both experimental setups (inhibition of biofilm formation and eradication of formed biofilm), a panel of the following controls was included: DMSO treated and not irradiated biofilm (+DMSO; -light), DMSO treated and irradiated biofilm (+DMSO; +light), PS treated and not irradiated biofilm (+PS; -light), untreated and irradiated biofilm (-PS; +light) and untreated and not irradiated biofilm (-PS; -light). All experiments were independently repeated at least three times.

### 4.7. Confocal Microscopy Analyses

Anti-biofilm activity of porphyrins on *Pseudomonas aeruginosa* PAO1 was analyzed using PAO1_pVOGFP recombinant strain, in which GFP fluorescent protein is expressed under the control of pBAD arabinose inducible promoter [51]. Overnight culture of *P. aeruginosa* PAO1_pVOGFP was diluted 500-fold in M9 minimal medium added with glucose (10 mM) and casamino acids (0.2% *V*/*V*) reaching a concentration of ~10^7^ CFU/mL and inoculated on coverslip glass positioned in 35 mm Petri dish. P11 30 µM was added at a final concentration of 30 µM and incubated in the dark for 1 h. Upon irradiation with a final dose of 30 J/cm^2^ (100 mW/cm^2^, 300 s), bacteria were grown O/N at 37 °C to form biofilm. Planktonic phase was removed and GFP expression was induced for 1 h at 37 °C by the addition of fresh medium containing arabinose 0.1% *W*/*V*. Finally, the coverslip was placed on a microscope glass slide for the acquisition of the adherent biofilm images. To evaluate the eradication, the recombinant strain was inoculated as previously described in this section and let form biofilm on coverslip glass. A 24 h old biofilm, after 1 h dark incubation with P11 at 30 µM, was irradiated (30 J/cm^2^). After irradiation, GFP expression was induced for confocal analysis.

All microscopic image acquisitions were performed on a Leica TCS SP5 CLSM (Leica Microsystems, Wetzlar, Germany) equipped for GFP visualization (excitation laser at 488 nm). Images were obtained using a x63 objective lens. Simulated 3D images of *P. aeruginosa* biofilm were generated using the free open-source software ImageJ (National Institute of Health, Bethesda, MD, USA).

### 4.8. Photosensitizer Binding Assay

All the photosensitizers were tested for their ability to bind bacterial cells. Upon overnight growth of *P. aeruginosa* PAO1, cultures were centrifuged at 5000× *g* for 10 min and the supernatants were removed. Pellets were resuspended and 10-fold diluted in sterile deionized water to obtain samples at 10^8^ CFU/mL. The bacterial concentration was evaluated by a plate count technique. Porphyrins at the concentration of 30 µM were added to the cells and samples were incubated for 1 h at 37 °C in the dark. This concentration was optimal to detect absorbance spectrum of all the tested porphyrins. Untreated cells, PSs treated cells and cells added with DMSO 4% (*V*/*V*) were included as controls. After dark incubation, samples were centrifuged (10,000× *g* for 5 min) and the visible spectra of the supernatants were recorded (k = 380–700 nm). A calibration plot (µM vs. OD) was obtained for each PS. The amount of PS not bound to bacterial cells was inferred interpolating the data on the calibration plot. The percentage of each PS bound to *P. aeruginosa* cells is represented as the mean ± standard deviation of at least three independent experiments.

### 4.9. Statistical Analyses

Photoinactivation experiments on suspended cells and biofilm formation by each microbial strain were performed at least three times with independent cultures, and statistical analyses were assessed by one-way ANOVA. If homogeneity of variance was not observed, post hoc test was performed.

## Figures and Tables

**Figure 1 ijms-22-06808-f001:**
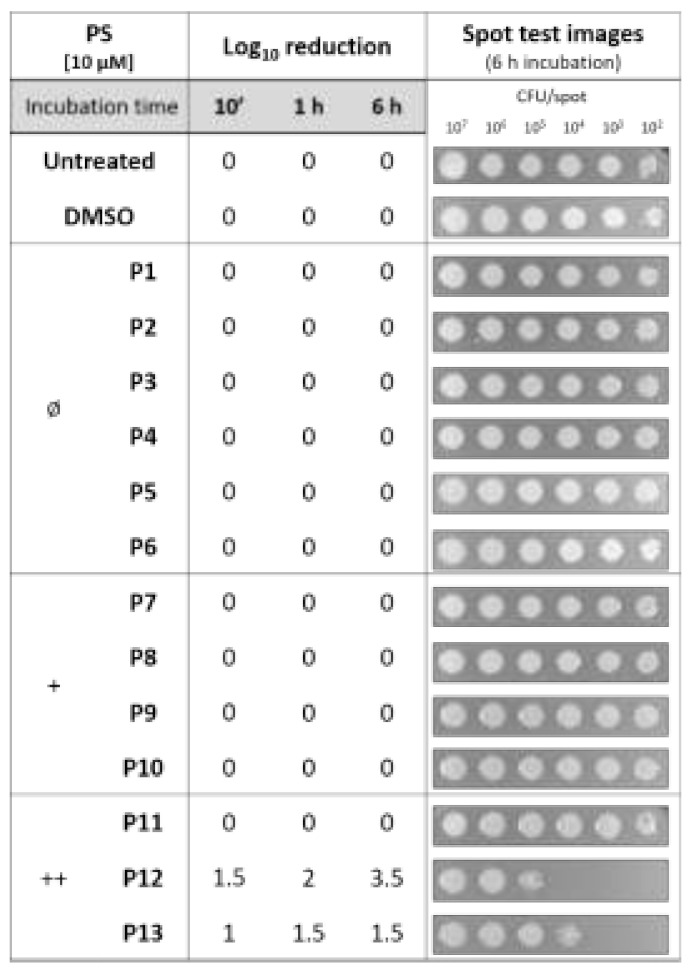
Analysis of intrinsic toxicity of diaryl-porphyrins (P1-P13). PSs were administered at a concentration of 10 µM to samples of *P. aeruginosa* PAO1 at decreasing concentrations (from 10^9^ up 10^4^ CFU/mL). After dark incubation for 10 min, 1 and 6 h, volumes of ~5 µL of each sample were replica plated on LB agar. After overnight incubation at 37 °C, the growth spots were checked. In the second column, Log10 reduction values represent the mean of three independent experiments for each dark incubation. Representative images reported in the last column refer to growth spots at the corresponding bacterial denisities (from 10^7^ to 10^2^ CFU/spot) upon 6 h of dark incubation with the tested PS. Each experiment has been repeated at least three times with independent cultures.

**Figure 2 ijms-22-06808-f002:**
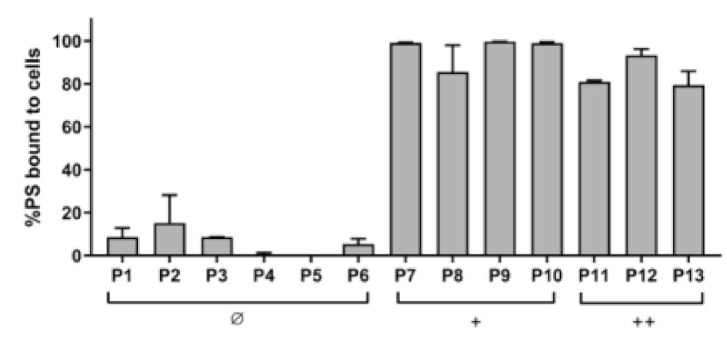
Binding assay of diaryl-porphyrins (P1–P13) to *P. aeruginosa* PAO1. Bacterial cells at 10^8^ CFU/mL were dark incubated for 1 h with neutral (Ø), monocationic (+) and dicationic (++) porphyrins administered at a concentration of 30 μM. Samples were centrifuged (10,000× *g*, 5 min) and the visible spectra of the supernatants were recorded. The rate of bound PS was inferred. Three independent experiments have been performed and the mean ± standard deviation of percentage of PS bound to cells is reported for each diaryl-porphyrin.

**Figure 3 ijms-22-06808-f003:**
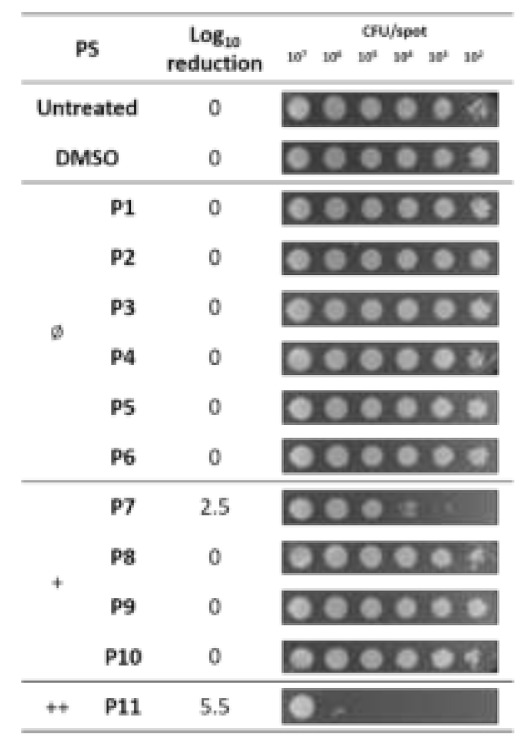
Photodynamic activity of diaryl-porphyrins on *P. aeruginosa* PAO1 evaluated by the photo-spot test. The neutral (Ø), monocationic (+) and dicationic (++) porphyrins were administered at a final concentration of 10 μM to samples of *P. aeruginosa* PAO1 at decreasing concentrations (from 10^9^ up 10^4^ CFU/mL). After 1 h dark incubation, volumes of ~5 µL of each sample were replica plated on LB agar and irradiated under 410 nm light (20 J/cm^2^). Cells were incubated at 37 °C O/N and growth spots were checked and representative images are reported in the last column. Log10 reduction values represent the mean of at least three independent experiments.

**Figure 4 ijms-22-06808-f004:**
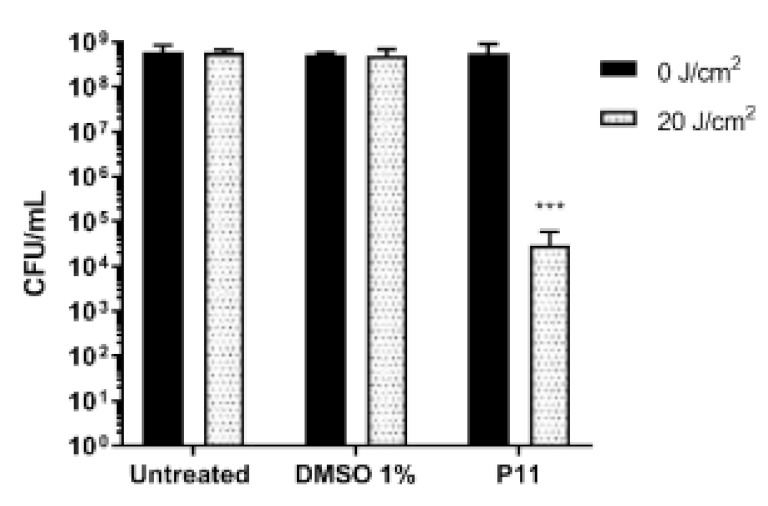
Photodynamic inactivation of *P. aeruginosa* PAO1 by P11. Bacterial samples at ~10^8^ CFU/mL suspended in water were incubated in the dark for 1 h with P11 10 µM. After dark incubation, cells were irradiated under light at 410 nm (20 J/cm^2^) and cellular viability was checked. Values, presented as CFU/mL, are the mean of at least three independent experiments and the bars represent standard deviations. Statistical analyses were performed by one-way ANOVA *** *p* < 0.0001).

**Figure 5 ijms-22-06808-f005:**
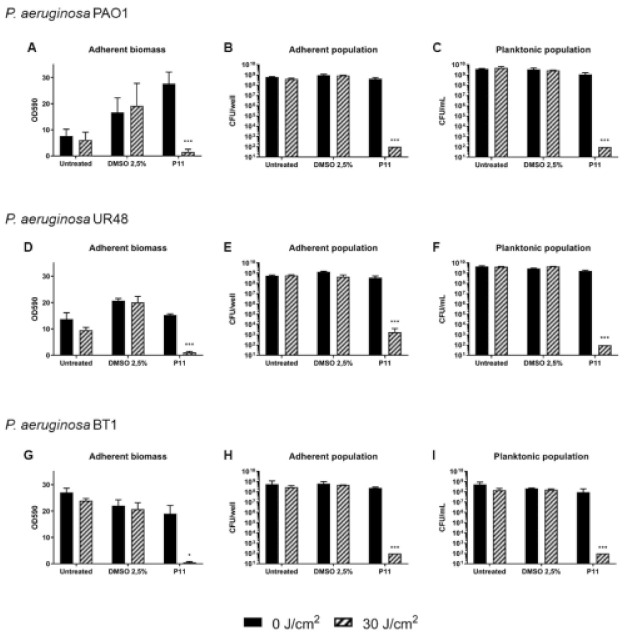
Inhibition of biofilm formation of *P. aeruginosa* PAO1 (**A**–**C**), UR48 (**D**–**F**) and BT1 (**G**–**I**) upon photodynamic treatment with diaryl-porphyrin P11. Overnight cultures of *P. aeruginosa* PAO1, UR48 and BT1 strains were diluted 500-fold in M9 minimal medium added with glucose (10 mM) and casamino acids (0.2% *V*/*V*) reaching a concentration of ~10^7^ CFU/mL and inoculated in 12-well microplate. P11 was added at a final concentration of 30 µM and incubated in the dark for 1 h. Upon irradiation with a final dose of 30 J/cm**^2^** (100 mW/cm^2^, 300 s), bacteria were grown O/N at 37 °C to form biofilm. The graphs report values of the optical density at 590 nm (OD 590) after biofilm staining with crystal violet (**A**,**D**,**G**), values of adherent population density (CFU/well) (**B**,**E**,**H**) and planktonic population concentration (CFU/mL) (**C**,**F**,**I**). Dark control samples are represented as black bars and light-treated samples as striped bars. Data represent the mean of at least three independent experiments ± the standard deviation. Statistical analyses were performed by one-way ANOVA (* *p* < 0.05; *** *p* < 0.0001).

**Figure 6 ijms-22-06808-f006:**
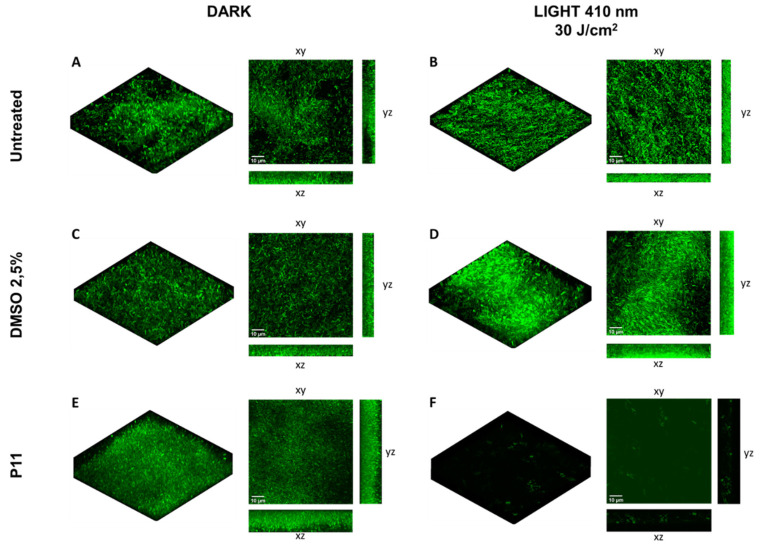
Inhibition of biofilm formation by *P. aeruginosa* PAO1-GFP by dicationic diaryl-porphyrin P11. GFP-tagged PAO1 cells were inoculated on coverslip glass and treated with P11 30 µM for 1 h in the dark. Cells were irradiated with light at 410 nm (30 J/cm^2^) and incubated at 37 °C in static to let form biofilm. After overnight incubation, GFP expression was induced and biofilm formed on glasses was analyzed by confocal microscopy. Untreated and DMSO-treated biofilms are included as controls. Confocal images of dark controls are shown in panels (**A**,**C**,**E**), while irradiated samples are shown in panels (**B**,**D**,**F**). Images of biofilms are shown in volume view, and in xy, xz and yz projections (scale bar = 10 µm).

**Figure 7 ijms-22-06808-f007:**
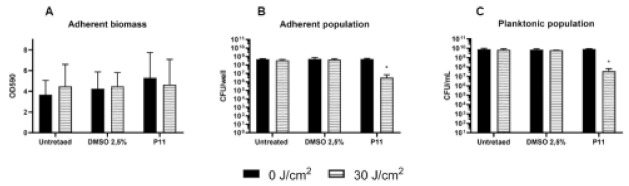
Assay of eradication of *P. aeruginosa* PAO1 biofilm by porphyrin P11. Overnight cultures of *P. aeruginosa* PAO1 were diluted 500-fold in M9 minimal medium added with glucose (10 mM) and casamino acids (0.2% *V*/*V*) reaching a concentration of ~10^7^ CFU/mL and inoculated in 12-well microplate to let form biofilm. 24 h-old biofilm was treated with PS at a concentration of 30 µM. After dark incubation for 1 h, biofilm was irradiated (30 J/cm^2^). After irradiation, the adherent biomass (OD_590_) (**A**) and viable counts from sessile (CFU/well) (**B**) and planktonic phases (CFU/mL) (**C**), respectively, have been evaluated. Data represent the mean of at least three independent experiments ± the standard deviation. Statistical analyses were performed by one-way ANOVA (* *p* < 0.05).

**Figure 8 ijms-22-06808-f008:**
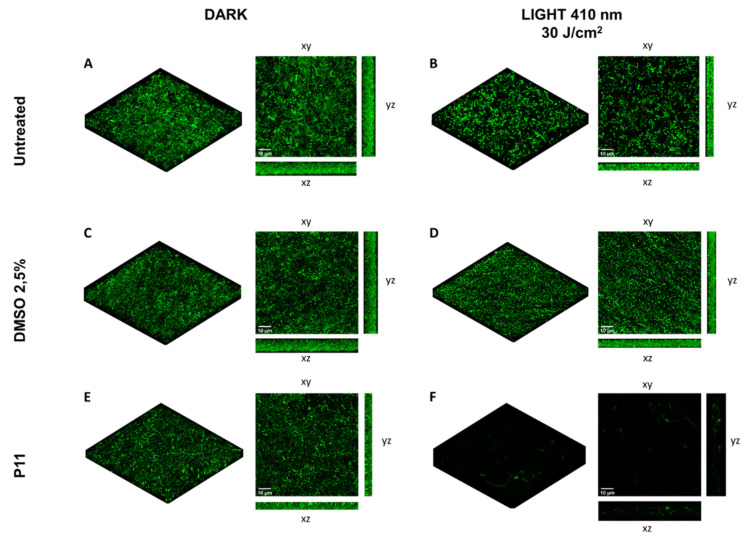
Assay of *P. aeruginosa* PAO1-GFP biofilm eradication with dicationic diaryl-porphyrin P11. A 24 h-old PAO1-GFP biofilm grown on coverslip glass was treated with P11 (30 µM final concentration) and irradiated with 410 nm blue light at 30 J/cm^2^. Upon induction of GFP expression, confocal analyses have been performed. Biofilms (24 h) of untreated and DMSO-treated samples were included in the experiment. Dark controls are shown in panels (**A**,**C**,**E**), while irradiated samples are depicted in panels (**B**,**D**,**F**). Images of biofilms are shown in volume view, and in xy, xz and yz projections (scale bar = 10 µm).

**Figure 9 ijms-22-06808-f009:**
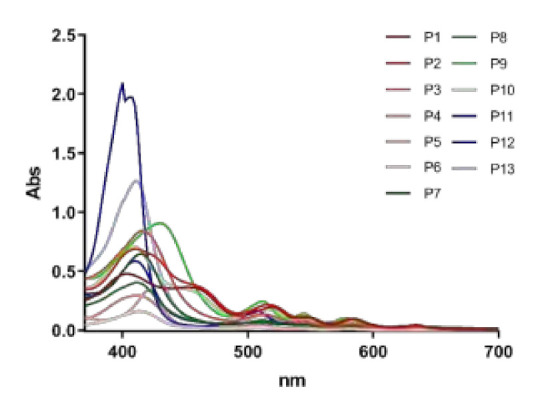
Visible light absorption spectra of diaryl-porphyrins (P1-P13).

**Table 1 ijms-22-06808-t001:** List of diaryl-porphyrins (P1-P13) used in this study.

PS		Chemical Structure	Chemical Denomination	Ref
Non-ionic (ø)	P1	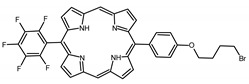	5-Pentafluorophenyl-15-[4-(4-Bromobutoxy)Phenyl]-21H,23H-porphyrin	[34]
	P2	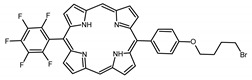	5-Pentafluorophenyl-15-[4-(8-Bromooctanoxy)Phenyl]-21H,23H-porphyrin	[34]
	P3	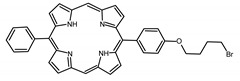	5-Phenyl-15-[4-(4-bromobutoxy)phenyl]-21H,23H-porphyrin	[31]
	P4	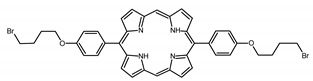	5,15-Di[4-(4-bromobutoxy)phenyl]-21H,23H-porphyrin	[31]
	P5	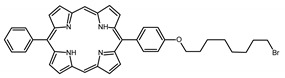	5-Phenyl-15-[4-(8-bromooctanoxy)phenyl]-21H,23H-porphyrin	[31]
	P6	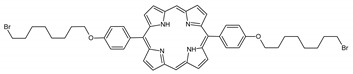	5,15-Di[4-(8-bromooctanoxy)phenyl]-21H,23H-porphyrin	[31]
Monocationic (+)	P7	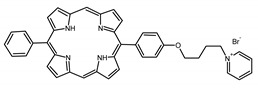	5-Phenyl-15-[4-(4-pyridinobutoxy)phenyl]-21H,23H-porphyrin	[31]
	P8	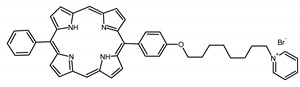	5-Phenyl-15-[4-(4-pyridinooctanoxy)phenyl]-21H,23H-porphyrin	[31]
	P9	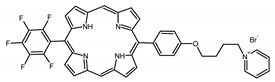	5-Pentafluorophenyl-15-[4-(4-Pyridinobutoxy)Phenyl]-21H,23H-porphyrin	[34]
	P10	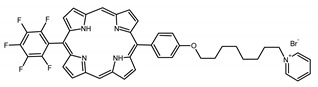	5-Pentafluorophenyl-15-[4-(4-Pyridinooctanoxy)Phenyl]-21H,23H-porphyrin	[34]
Dicationic (++)	P11	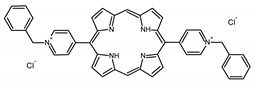	5,15-di(N-benzyl-4-pyridyl)-21H,23H-porphyrin	[33]
	P12	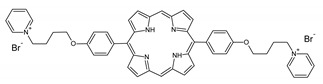	5,15-Di[4-(4-pyridinobutoxy)phenyl]-21H,23H-porphyrin	[31]
	P13	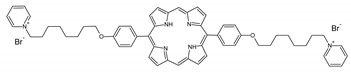	5,15-Di[4-(4-pyridinooctanoxy)phenyl]-21H,23H-porphyrin	[31]

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
