# Peer review of "Photoinactivation of Pseudomonas aeruginosa Biofilm by Dicationic Diaryl-Porphyrin"

_ijms, 2021, doi:10.3390/ijms22136808_

Round 1

Reviewer 1 Report

Line 16- please define “PS” since this is the first time you use this term.

Line 16-17 - Did you find out that two dicationic compounds are toxic after you have finished all the experiments in this study or did you know that they are toxic before this study?

Line 20 - please define “PAO1” since this is the first time you use this term.

Line 31 - The word “Thus” seems unnecessary here. 

Line 28-29 - Original reference is not mentioned.

Line 31-32 - Original reference is not mentioned

Line 40-42 - Original references are missing here

Line 42-44 - Original references are missing here

Many other human pathogens form biofilm. But authors haven’t explained why P. Aeruginosa biofilms are given special attention compared to other human pathogens. The briefly mention about multidrug resistant strains but it needs to be explained clearly.

Line 61-63 - This sentence might need revising. Not sure what authors meant by “Add value”

Line 63-64 - When authors say “several studies highlight the potential” they might want to provide the references for “several” instead of just one reference.

It would be better if authors included a section on toxicity of diaryl-porphyrins in the introduction.

Table 1 - chemical structures are not clear. Pls use high resolution figures.

Details of bacteria growth conditions (Temperature, concentration, media, source, etc) are missing in the materials and methods. 

2.1.1 Intrinsic toxicity of diaryl-porphyrins - What is the initial bacteria concentration? The overnight growth of bacteria in two different experiments may result in different number of bacteria. Therefore, the results you see in two different experiments may not be the same. How did you make sure that you started your experiments with the same number of bacteria. ie. How did you calculate/determine the bacteria concentration in your overnight culture?

Why did intrinsic toxicity of a compound to PAO1 made those compounds unsuitable as an antimicrobial?

Figure 1 - please include details of the experiment in the figure caption. ie. Number of replicates per treatment, number of times the experiment was repeated, etc.

2.2. Diaryl-porphyrins binding and photoinactivation rates - Pls refer to the details and questions for the 2.1.1. section.

How did you remove, treatments from PAO1 cells before measuring %PS binding?

Line 152 - Authors mention “the dark toxicity is not desired in aPDT applications”. Please explain why dark toxicity is not desired. 

Figure 3- Please see the comments for figure 1,2.

How did you select 10uM (30 uM for biofilm assay) concentration as treatment concentration?

Overnight incubation of plates after treatment may not provide complete results. Did you check plates for colony growth beyond overnight incubation?

Line 471-472 - It looks like authors didn’t wash cells after porphyrin treatment during the test for cell binding of porphyrins. Therefore, the results of binding experiment is questionable since there is no way to determine bound vs suspended porphyrin.

Line 474 - Materials and methods section of the Photosensitizer binding assay mention about yeast cells. But there was no results for yeast anywhere in the article.

Figure 5 - The resolution of the figure is very low and it is very difficult to read. Please use high resolution figures.  

Please include details of biofilm quantification in materials and methods (the details in the Fig 5 caption for biofilm quantification may not be enough.) Did you do the experiment in test tubes or microtiter plates? How did you measure OD reading (ie. Directly from the 96 well plate or otherwise? Did you do ethanol washing after crystal violet staining?

Please indicates the starting number of bacteria cells for this experiment and how did you determine it (ie. McFarland standard, OD600 etc.)

How did you calculate CFU values in the graph (Fig 5) for the adherent population?

Authors have not provided enough data to support that P11 is a good candidate to treat P. aeruginosa infections. They haven’t done any mamalian cell toxicity studies at all. Without providing exact effective concentration of p11 and the study does provide any new knowledge. At minimum they must have provided, minimum inhibitory concentration (bacteriostatic concentrations or bactericidal concentration together with highest possible p11 concentration  without any mamalian cell toxicity.

The normal procedure to test effectiveness of antimicrobial compounds is minimum inhibitory concentration (MIC) assay with a standard initial bacteria concentration (5x105 cells). For more details please refer to the following article (Wiegand et al 2008). Based on your experimental results, readers can see which of your test material perform better compared to each other, but there is no way to compare antimicrobial effects of PS compounds compared to other material available or published results, because the authors haven’t used any standard tests (for example MIC test). Authors could have used a positive control (another known antimicrobial compound) for comparison (it doesn’t have to be photoactive). 

The materials and methods section of the biofilm measurements does not provide full details to understand the experimental procedure (ie. Whether the experiments were done in test tubes, 96 well plates etc.). Therefore it is very difficult to evaluate the experiments.

Wiegand I, Hilpert K, Hancock RE. Agar and broth dilution methods to determine the minimal inhibitory concentration (MIC) of antimicrobial substances. Nature protocols. 2008;3(2):163. pmid:18274517

Author Response

1) Line 16- please define “PS” since this is the first time you use this term.

The definition has been added.

2) Line 16-17 - Did you find out that two dicationic compounds are toxic after you have finished all the experiments in this study or did you know that they are toxic before this study?

The authors emphasized that the intrinsic toxicity of two dicationic PSs was highlighted in the present study.

3) Line 20 - please define “PAO1” since this is the first time you use this term.

As suggested by referee, the species Pseudomonas aeruginosa has been added to define the strain PAO1.

4) Line 31 - The word “Thus” seems unnecessary here. 

The adverb was removed.

5) Line 28-29 - Original reference is not mentioned.

The following original reference has been mentioned:

“Rice, L.B. Federal Funding for the Study of Antimicrobial Resistance in Nosocomial Pathogens: No ESKAPE. 1094–102, doi:10.1086/533452.”

6) Line 31-32 - Original reference is not mentioned

The following reference has been mentioned:

“Ma YX, Wang CY, Li YY, Li J, Wan QQ, Chen JH, Tay FR, Niu LN. Considerations and Caveats in Combating ESKAPE Pathogens against Nosocomial Infections. Adv Sci (Weinh). 2019 Dec 5;7(1):1901872. doi: 10.1002/advs.201901872.

7) Line 40-42 - Original references are missing here

The following references have been mentioned:

  • Nordmann, P.; Naas, T.; Fortineau, N.; Poirel, L. Superbugs in the coming new decade; multidrug resistance and prospects for treatment of Staphylococcus aureus, Enterococcus spp. and Pseudomonas aeruginosa in 2010. Opin. Microbiol. 2007, 10, 436–440.
  • Migiyama, Y.; Yanagihara, K.; Kaku, N.; Harada, Y.; Yamada, K.; Nagaoka, K.; Morinaga, Y.; Akamatsu, N.; Matsuda, J.; Izumikawa, K.; et al. Pseudomonas aeruginosa bacteremia among immunocompetent and immunocompromised patients: Relation to initialantibiotic therapy and survival. J. Infect. Dis. 2016, 69, 91–96, doi:10.7883/yoken.JJID.2014.573.

8) Line 42-44 - Original references are missing here:

The following reference have been mentioned:

  • Driscoll, J.A.; Brody, S.L.; Kollef, M.H. The epidemiology, pathogenesis and treatment of Pseudomonas aeruginosa infections. Drugs 2007, 67, 351–368, doi:10.2165/00003495-200767030-00003.
  • El Zowalaty, M.E.; Al Thani, A.A.; Webster, T.J.; El Zowalaty, A.E.; Schweizer, H.P.; Nasrallah, G.K.; Marei, H.E.; Ashour, H.M. Pseudomonas aeruginosa: Arsenal of resistance mechanisms, decades of changing resistance profiles, and future antimicrobial therapies. Future Microbiol. 2015, 10, 1683–1706, doi:10.2217/fmb.15.48.

9) Many other human pathogens form biofilm. But authors haven’t explained why P. Aeruginosa biofilms are given special attention compared to other human pathogens. The briefly mention about multidrug resistant strains but it needs to be explained clearly.

As observed by referee, in natural environment, most microbial species could form biofilm. In this study, the authors focused their attention on Pseudomonas aeruginosa that belongs to “Eskape” pathogen group (lines #28-31). P. aeruginosa is an opportunistic pathogen and the biofilm formation represents an important virulence factor. As suggested by referee, a more detailed description of the role of biofilm for P. aeruginosa has been added (lines # 45-60), as follows:

“The major cause of persistent P. aeruginosa infections is the presence of biofilm, formed on tissues or on the surface of surgical implants and medical devices. P. aeruginosa cells attach on a surface through twitching motility driven by type IV pili and develop microcolonies. At this step, bacteria secrete exopolysaccharides (Psl, Pel and alginate) and other components such as polypeptides and extracellular DNA. During biofilm maturation, the community acquires its typical three-dimensional structure that planktonic cells can leave for colonizing other surfaces [6]. Biofilm formation and the production of many virulence factors (i.e. pyocyanin, pyoverdine, elastases, proteases, rhamnolipids, exotoxin A) contribute to pathogenicity in P. aeruginosa [7]. Furthermore, P. aeruginosa is characterized by an outer membrane that acts as a selective barrier to prevent the antibiotic entrance, in addition to several non -specific porins that govern membrane permeability [8]. In addition, a wide variety of efflux pumps (i.e. mexAB-OprM, MexCD-OprJ, MexEF-OprN) are responsible for the resistance to different classes of antibiotics (β-lactams, quinolones and aminoglycosides) [9]. The production of extended-spectrum β -lactamases and enzymes modifying aminoglycosides worsen the spread of multidrug resistant strains [10].“

10) Line 61-63 - This sentence might need revising. Not sure what authors meant by “Add value”.

The sentence has been changed as follows:” An interesting advantage of this technique is based on the efficacy observed against both sensitive and antibiotic resistant strains”

11) Line 63-64 - When authors say “several studies highlight the potential” they might want to provide the references for “several” instead of just one reference.

The following references have been added:

- Pithan, E.; Righi, M.; Bruno, C.; Christ, R.; Antonio, M.; Zanini, K. Photodiagnosis and Photodynamic Therapy Antimicrobial photodynamic effect of phenothiazinic photosensitizers in formulations with ethanol on Pseudomonas aeruginosa biofilms. Photodiagnosis Photodyn. Ther. 2016, 13, 291–296, doi:10.1016/j.pdpdt.2015.08.008.

- Abdulrahman, H.; Misba, L.; Ahmad, S.; Khan, A.U. Curcumin induced photodynamic therapy mediated suppression of quorum sensing pathway of Pseudomonas aeruginosa: An approach to inhibit biofilm in vitro. Photodiagnosis Photodyn. Ther. 2020, doi:10.1016/j.pdpdt.2019.101645.

- Sarker, R.R.; Tsunoi, Y.; Haruyama, Y.; Ichiki, Y.; Sato, S.; Nishidate, I. Combined Addition of Ethanol and Ethylenediaminetetraacetic Acid Enhances Antibacterial and Antibiofilm Effects in Methylene Blue-Mediated Photodynamic Treatment against Pseudomonas aeruginosa In Vitro. Photochem. Photobiol. 2021, 97, 600–606, doi:10.1111/php.13358.

- Anju, V.; Paramanantham, P.; Siddhardha, B.; Sruthil Lal, S.; Sharan, A.; Alyousef, A.A.; Arshad, M.; Syed, A. Malachite green-conjugated multi-walled carbon nanotubes potentiate antimicrobial photodynamic inactivation of planktonic cells and biofilms of Pseudomonas aeruginosa and Staphylococcus aureus. 2019, doi:10.2147/IJN.S202734.

12) It would be better if authors included a section on toxicity of diaryl-porphyrins in the introduction.

As suggested by referee, the authors enriched the introduction with the following paragraph on toxicity that has been moved from discussion section.

Lines # 99-108: “Photosensitizers should not be toxic and mutagenic in the dark towards both eukaryotic and prokaryotic cells. In literature, conflicting observations have been reported on dark toxicity of the most investigated porphyrin, tetracationic TMPyP (5,10,15,20-Tetrakis(1-methyl-4-pyridinio)-porphyrin tetra(p-toluenesulfonate). Heckl assumed that the dark toxicity of TMPyP, observed in several bacterial species, could be attributed to photoinactivation effects caused by any residual light in a laboratory. When keeping the bacteria under dark conditions (< 10 nW cm²), no dark toxicity in Escherichia coli was detected for a very high concentration of TMPyP (250 mM) and incubation times up to 24 h [29]. On the other hand, P. aeruginosa strains isolated from Fibrosis Cystic patients were sensitive in the dark to porphyrins [30].

13) Table 1 - chemical structures are not clear. Pls use high resolution figures.

Since the table 1 represents many chemical structures, it is not possible to increase the resolution of figures. To make clearer the chemical structure of porphyrins represented in table 1, the lines in the present version are thicker than original.

14) Details of bacteria growth conditions (Temperature, concentration, media, source, etc) are missing in the materials and methods.

In material and method section, “Microbial strains and culture conditions” paragraph describes the bacterial growth conditions. In this section, the method used to evaluate cellular concentration has been added.

15) 2.1.1 Intrinsic toxicity of diaryl-porphyrins –

What is the initial bacteria concentration? The overnight growth of bacteria in two different experiments may result in different number of bacteria. Therefore, the results you see in two different experiments may not be the same. How did you make sure that you started your experiments with the same number of bacteria. ie. How did you calculate/determine the bacteria concentration in your overnight culture?

  • The concentration of bacterial culture, after overnight growth in Luria Bertani (LB) medium on an orbital shaker at 200 rpm at 37°C, is ⁓ 109 CFU/ml. This is the start point for all the described experiments. The cellular concentration reached upon O/N growth has been added to make clearer the protocols.
  • The bacterial concentration, upon overnight growth, is checked through an “indirect” method. A calibration plot, previously obtained, correlates the optical density at 600 nm of different samples of aeruginosa PAO1 suspended in LB with cellular concentration by means of plate count technique. The spectrophotometer measurement (OD600) of overnight cultures is used to interpolate the corresponding bacterial concentration expressed as CFU/mL from calibration plot. In the chosen experimental conditions, the growth of PAO1 in 20 mL of LB at 37°C in Enlermayer flask of 100 mL is ⁓ 109 CFU/mL.

16) Why did intrinsic toxicity of a compound to PAO1 made those compounds unsuitable as an antimicrobial?

In antimicrobial photodynamic approach, the ideal PS should not be intrinsically toxic. The reduction of microbial viability, in the presence of the photosensitizer without irradiation, is considered as dark toxicity. The PS toxicity should be selective for microbial cells upon its photoactivation. We should be able to choose when activate PSs, irradiating the target and killing microorganisms.

This message has been emphasized by adding the following sentence (lines #136-137): “The reduction of microbial viability in the presence of photosensitizer without irradiation should be avoided.”

17) Figure 1 - please include details of the experiment in the figure caption. ie. Number of replicates per treatment, number of times the experiment was repeated, etc.

The caption of figure 1 has been enriched with further experimental details as suggested by referee.

18) 2.2. Diaryl-porphyrins binding and photoinactivation rates - Pls refer to the details and questions for the 2.1.1. section.

Authors appreciate the suggestion, and more details have been added to binding protocol section. The caption of figure 2 has been enriched with experimental details.

19) How did you remove, treatments from PAO1 cells before measuring %PS binding?

Bacterial cells were incubated with PSs 30 µM in the dark for 1 h. After dark incubation, cells were centrifuged (10000 g for 5 min). The supernatant was analysed to detect the amount of PS. Since author’s aim was to evaluate the rate of PS able to interact both strongly and/or weakly with cell envelope, no further step of washing has been performed. As can be appreciated by figure 2, this protocol is useful to distinguish the opposite interaction of cationic and neutral compounds. 

20) Line 152 - Authors mention “the dark toxicity is not desired in aPDT applications”. Please explain why dark toxicity is not desired

As previously reported, the following sentence has been added: “The reduction of microbial viability in the presence of photosensitizer without irradiation should be avoided.”

21) Figure 3- Please see the comments for figure 1,2.

The caption of figure 3 has been enriched with experimental details as suggested by referee.

22) How did you select 10 µM (30 µM for biofilm assay) concentration as treatment concentration?

In antimicrobial photodynamic experiments, the best of author knowledge, PSs are administered to bacteria in the range of micromolar. To photoinactivate P. aeruginosa, known to be tolerant to photoinactivation, 10 mM was optimal for the initial screening of diarylporphyrins. However, the administration of P11 10 mM to suspended cells did not cause a good killing rate. Consequently, to inhibit and/or eradicate P. aeruginosa biofilm, it was necessary to administer a higher PS concentration (30 mM). This concentration was not toxic in the dark to P. aeruginosa cells.

23) Overnight incubation of plates after treatment may not provide complete results. Did you check plates for colony growth beyond overnight incubation?

The observation of referee is interesting: incubation longer than 24 hours could reduce the difference between treated (+PS, + light) and untreated samples (-PS, -light). However, authors decided to check the spot after 24 hours, a time necessary to control samples (-PS, -light) to reach a visible growth. The growth delay can be considered an interesting goal for in in vivo applications (disinfection and sanitization).

24) Line 471-472 - It looks like authors didn’t wash cells after porphyrin treatment during the test for cell binding of porphyrins. Therefore, the results of binding experiment is questionable since there is no way to determine bound vs suspended porphyrin.

As previously described, author’s aim was to evaluate the rate of PS able to interact both weakly and/or strongly with cell envelope. The porphyrins suspended in the supernatant and detected with spectrophotometric measurement are molecules that did not bind to cells. The only drawback of this protocol is that it is not possible distinguish between bound PS and up-taken PS. However, this approach highlights the different interaction of cationic and neutral porphyrins: the first more prone to interact than the latter.

25) Line 474 - Materials and methods section of the Photosensitizer binding assay mention about yeast cells. But there was no results for yeast anywhere in the article.

The authors amended the error.

26) Figure 5 - The resolution of the figure is very low and it is very difficult to read. Please use high resolution figures.

As suggested by the referee, the authors improved the resolution of Figure 5 from 700 dpi to 1200 dpi.

27) Please include details of biofilm quantification in materials and methods (the details in the Fig 5 caption for biofilm quantification may not be enough.) Did you do the experiment in test tubes or microtiter plates? How did you measure OD reading (ie. Directly from the 96 well plate or otherwise? Did you do ethanol washing after crystal violet staining?

The comment of the referee is relevant. A deeper and detailed description of biofilm quantification has been added in material and method section.

28) Please indicates the starting number of bacteria cells for this experiment and how did you determine it (ie. McFarland standard, OD600 etc.)

The cellular concentration has been performed by a plate count technique. This procedure has been described at the beginning of material and method section (lines #425-431).

29) How did you calculate CFU values in the graph (Fig 5) for the adherent population?

The procedure has been described, as suggested by referee, in material and method section.

30) Authors have not provided enough data to support that P11 is a good candidate to treat P. aeruginosa infections. They haven’t done any mammalian cell toxicity studies at all. Without providing exact effective concentration of p11 and the study does provide any new knowledge. At minimum they must have provided, minimum inhibitory concentration (bacteriostatic concentrations or bactericidal concentration together with highest possible p11 concentration without any mammalian cell toxicity.

The authors did not show any results on mammalian cells. Indeed, the goal of this study was to compare the anti-Pseudomonas activity of diarylporphyrins sharing a very similar structure. In literature, it is not easy to find manuscripts reporting the comparison of compounds tested in the same experimental set-up. The reader is guided along an ideal journey that could be used to choose the best antimicrobial PS. The authors applied the same flow to test diaryl-porphyrins in Candida albicans (Cosmetics 2020, 7, doi:10.3390/COSMETICS7020031). Further studies are in progress with other compounds and other microorganisms. At the end, the best candidates will be tested to rule out host toxicity.

31) The normal procedure to test effectiveness of antimicrobial compounds is minimum inhibitory concentration (MIC) assay with a standard initial bacteria concentration (5x105 cells). For more details please refer to the following article (Wiegand et al 2008). Based on your experimental results, readers can see which of your test material perform better compared to each other, but there is no way to compare antimicrobial effects of PS compounds compared to other material available or published results, because the authors haven’t used any standard tests (for example MIC test). Authors could have used a positive control (another known antimicrobial compound) for comparison (it doesn’t have to be photoactive).

The consideration of referee is legitimate. Nevertheless, the guidelines declared by Eucast or other International Committee cover antimicrobial drugs and/or antibiotics with action mechanisms different from photosensitizers. In photodynamic world, dyes are comparable to disinfectants for which common guidelines are not available. The dark incubation, the light source and energy parameters complicate the standardization. Hopefully, in the future, a standardized protocol could be shared by photodynamic community in order to reach the desired goal. The optimized protocols aimed at calculating MIC, as cited in the suggested paper, is not satisfying for photodynamic applications. The aim of the present study is to perform a comparison of the photoactivity of members of the same family. The goal is to choose the best among them.

The authors added, in the discussion section, the suggested reference at line # 397.

32) The materials and methods section of the biofilm measurements does not provide full details to understand the experimental procedure (ie. Whether the experiments were done in test tubes, 96 well plates etc.). Therefore it is very difficult to evaluate the experiments.

More details have been added to materials and methods section to complete description for biofilm preparation and photoinactivation.

Reviewer 2 Report

The submitted manuscript is well written and structured. I find the discussion correct and the methodology well presented.

I think that the authors overuse bar graphs, although they are intuitive they should be accompanied by tables, at least in Figure 5, numerical data would help to see the differences between the three strains used.

On the other hand, the structures shown in Table 1 are unintelligible. I propose a Fig 1 with the structures of the panel of diaryl-phorphyrins tested and Table 1 including only the numbering, name and reference.

Author Response

1)I think that the authors overuse bar graphs, although they are intuitive they should be accompanied by tables, at least in Figure 5, numerical data would help to see the differences between the three strains used.

The authors improved the resolution of Figure 5, hoping that the improved image quality would make understanding the numerical data more intuitive.

2) On the other hand, the structures shown in Table 1 are unintelligible. I propose a Fig 1 with the structures of the panel of diaryl-phorphyrins tested and Table 1 including only the numbering, name and reference.

The authors considered the suggestion of referee and split figures from name and references of Table 1. The new format loses the convenience of table that links the name/acronym to chemical structures and permit to follow more easily the reading. In order to make clearer, the lines of chemical structures have been made thicker than previous ones.

Reviewer 3 Report

The proposed work by Orlandi and colleagues aimed to investigate the photoinactivation of biofilm of Pseudomonas aeruginosa.

The research is well conducted and provides to give other information about the defeat of this pathogen.

The research followed a very linear experimental design. I don't any comment related to research, but I encourage the English revision.

For this reason, I give a minor revision of the manuscript.

Author Response

The research followed a very linear experimental design. I don't any comment related to research, but I encourage the English revision.

As suggested by the referee, the work was revised trying to improve the quality of the English

Round 2

Reviewer 1 Report

Thank you for answering the questions I had and adding corrections as suggested. 

The manuscript quality is much better in current version!